# Rate of glycemic control and associated factors among type two diabetes mellitus patients in Ethiopia: A cross sectional study

Shambel Nigussie[1]*, Nigussie Birhan[2], Firehiwot Amare[1], Getnet Mengistu[3], Fuad Adem[1], Tadesse Melaku Abegaz[4,5]

1 Department of Clinical Pharmacy, School of Pharmacy, College of Health and Medical Science, Haramaya University, Harar, Ethiopia, 2 Department of Nursing, College of Medicine and Health Science, University of Gondar, Gondar, Ethiopia, 3 Pharmacology and Toxicology Unit, Pharmacy Department, College of Medicine and Health Science, Wollo University, Dessie, Ethiopia, 4 Department of Clinical Pharmacy, School of Pharmacy, College of Medicine and Health Science, University of Gondar, Gondar, Ethiopia, 5 Clinical and Health Science, University of South Australia, Adelaide, Australia

* shambelpharm02@gmail.com

**Data Availability Statement:** All relevant data are within the manuscript and its Supporting information files.

## Abstract

### Objective

To assess the rate of glycemic control and associated factors among type 2 diabetes mellitus patients at Dilchora Referral Hospital, Dire Dawa, Eastern Ethiopia.

### Methods

A cross-sectional study was conducted from 13 May to 16 August 2019. Type 2 diabetic patients on follow up at Dilchora Referral Hospital who fulfilled the inclusion criteria of the study were included. Systematic random sampling was used to select study participants. Data was collected by a face-to-face interview and review of medical records. The primary outcome was the level of blood glucose during three consecutive visits. Poor glycemic control was defined as a blood sugar level of more than 154 mg/dL based on the average of measurements from three consecutive visits. Multivariate logistic regression analysis was used to identify determinants of glycemic control.

### Result

A total of 394 participants responded to the interview and were included in the final analysis. The overall prevalence of poor glycemic control was 45.2% (95%CI: 40.6%-50.0%). Patients who were on oral anti-diabetic drug plus insulin had more than two times greater chance of poor glycemic control than patients on oral anti-diabetic drug alone: 2.177(95% CI:1.10–4.29). The odds of poor glycemic control in patients who did not understand the pharmacist's instructions was two times higher than patients with good understanding of instructions 1.86(95%CI: 1.10–3.13). Patients who had poor level of practice were found to have poor glycemic control: 1.69(95% CI: 1.13–2.55).

**Funding:** The authors received no specific funding for this study.

**Competing interests:** The authors have declared that no competing interests exist.

## Conclusion

The overall prevalence of poor glycemic control was high among type 2 diabetes patients. Oral anti-diabetic drugs in combination with insulin, lack of understanding of pharmacist's advice, and poor practice of diabetic patients were significant factors of poor glycemic control. Pharmacists should reassure the understanding of patients before discharge during counseling. Optimization of the dose of antidiabetic medications and combination of oral hypoglycemic agents should be considered.

## Introduction

Diabetes mellitus (DM) is a group of metabolic disorders which is characterized by the presence of high glucose level in the blood resulting from impairment in insulin secretion, insulin action, or both [1]. There are two broad types of DM, named type 1 which progresses as a result of autoimmunity against the insulin-producing beta cells and type 2 characterized by variable degrees of insulin resistance, impaired insulin secretion, and increased hepatic glucose production [2]. Globally, 451 million people were living with DM in 2017. This statistics was estimated to rise to 693 million by 2045 [3]. The global burden of disease data suggests DM to be responsible for 1.6 million deaths in 2016 [4, 5]. Despite the availability of a wide range of effective glucose-lowering therapies, approximately half of patients with type 2 diabetes mellitus (T2DM) in the world do not achieve glycemic targets. This increases the risk of diabetes-related complications and long-term health care costs [6, 7]. Poor blood glucose control causes about 7% of deaths among men aged 20–69 and 8% among female [8].

Maintaining blood sugar level within the range of ideal blood sugar control target is the most means of effective preventing complications associated with diabetes [9]. However, a ratio that is uncontrolled level of blood sugar for T2DM patients was very high. A multicenter study conducted in Eastern Europe, Asia, and Latin America showed that 96.4% of study participants had poor glycemic control [10]. Similarly, high proportions of T2DM patients with poor glycemic control ranging from 50% to 95.8% were reported in Brazil, south Indian, Karnataka, Uganda, Mthatha and Ghana [11–15]. In Ethiopia, hospital-based cross-sectional studies done at Gondar, Ambo, Jimma and Limmu indicated that 57.5%, 50%, 70.9%, 63.8% of participants had poor glycemic control, respectively [16–19]. The role of achieving the optimal blood glucose level in preventing the development and progression of complications is an established fact [9].

Even though studies established the influence of glycemic control on the progression of diabetic complications, small proportion of DM populations achieve the target glycemic level [11, 12, 20, 21]. Different factors have contributed to poor glycemic control including age, duration of the disease, type of treatment, patients' perception of health, educational level, occupation, and medication adherence [22–25]. However, there is limited evidence on how these factors are associated with poor glycemic control in Ethiopia [26, 27]. Additionally, the previous studies did not assess the influence of knowledge, attitude, and practice of diabetic patients and their interaction with health professionals particularly with pharmacists [25, 28, 29]. The present study aimed to assess poor glycemic control and associated factors in Dilchora Referral Hospital, Eastern Ethiopia.

## Methods and materials

### Study design, study area and study period

A cross-sectional study was conducted in Dilchora Referral Hospital, Eastern Ethiopia, from 13 May to 16 August 2019. Dilchora Referral Hospital is the only referral hospital in Dire-Dawa city administration that provides service to approximately 11229 inpatient and 118,886 outpatient attendees in 2016/2017 coming from a catchment population of 500,000.

### Source and study population

The source population was all DM patients who had follow-up at outpatient chronic follow-up of Dilchora Referral Hospital. The study population was all T2DM patients who had follow-up at Dilchora Referral Hospital during the study period who fulfilled the inclusion criteria of the study.

### Inclusion and exclusion criteria

T2DM patients who had follow-up at the clinic for at least one year with fasting blood glucose (FBG) measurements in the previous three consecutive months and above the age of 18 years were included. T2DM patients with mental disease, recurrent history of hypoglycemia and pregnant women were excluded.

### Study variables

The dependent variable was the rate of glycemic control and independent variables were sex, age, residence, educational level, occupation, marital status, religion, ethnicity, rate of interaction with pharmacist, clarity of pharmacist advice, patient language preference during interaction with pharmacist, overall satisfaction with pharmaceutical service, cholesterol level, body mass index (BMI), diabetic complication, comorbidity, duration of DM, type of medications, duration of treatment, knowledge, attitude and practice of diabetic patients.

### Sample size determination and sampling technique

The sample size was determined by using the formula of a single population proportion.

$$n = \frac{z^2 p(1-p)}{d^2}$$

where n is the sample size required; d, margin of error of 5% (d = 0.05); Z, the degree of accuracy required at 95% confidence level = 1.96; and P, prevalence rate of poor glycemic control. By reviewing different previous studies, we took a 50% prevalence rate of poor glycemic control which gave largest sample size and representativeness of the sample that finally ensures the generalization and precision of the findings [17].

Using the formula, the sample size was calculated as:

$$n = \frac{1.96^2 0.5(1-0.5)}{0.05^2} = 384$$

For possible nonresponse rate, 10% of the calculated sample was added to get a final sample size of 422 patients. Systematic random sampling technique was used to select study participants. The total numbers of T2DM patients who had follow-up at the diabetic clinic were 1308. The sampling interval was determined by dividing the number of patients on follow-up by the sample size of the study. The first patient was selected by lottery method from the list

prepared based on their medical record number and then every three patients was recruited into the study.

## Data collection methods

**Data collection tool.** A data abstraction format was used to record the necessary information from patients' medical records and a structured questionnaire was used to interview patients. By using the data abstraction format information on body mass index, comorbid disease, diabetic complication, type of antidiabetic treatment, blood sugar level, systolic blood pressure, diastolic blood pressure, and cholesterol levels was retrieved from patient's medical record. The structured questionnaire was prepared in English language and then translated to the local language (Amharic) to collect the data through face-to-face interviews. The questionnaire was prepared to collect information on residence, educational level, marital status, occupation, religion, ethnicity, family history of DM, duration of DM, participant interaction with the pharmacist, clarity of the pharmacist advice, patient language preference during interaction with the pharmacist, participants' satisfaction with overall pharmaceutical service, and knowledge, attitude, and practice of DM patients. Knowledge of T2DM patients was assessed by using 12 general questions about diabetes. Each response was scored as "1" for the correct answer and "0" for an incorrect answer. Participants who correctly answered more than 50% of knowledge questions were considered as having adequate knowledge whereas those who scored less than 50% were considered as having inadequate knowledge. On the other hand, 8 attitude and 10 practice-related questions were included in the questionnaire. The responses to each question were scored as "1" for the correct answer and "0" for an incorrect answer. Participants who correctly answered more than 50% of attitude and practices assessing questions were considered as having good attitude and practices whereas those who scored less than 50% were considered as having poor attitude and practice, respectively.

**Data collection procedure.** The data was collected by three trained nurses who working in Dilchora Hospital. All the required laboratory values were taken from the patient medical record.

**Data quality control.** To ensure the quality of data, a pretest was done on 5% of the total sample. The pretest was conducted at chronic follow-up of Dilchora Referral Hospital on randomly selected T2DM patients to ensure the accuracy of the data abstraction format and the structured questionnaire. The findings of the pretest were not included in the final analysis. Any error found during the process of the pretest was corrected and modification was made into the final version of the data abstraction format and the structured questionnaire. After developing the questionnaire by reviewing different literature; submitted to the experts to comment the questionnaire and by incorporating their comment finalized our questionnaire to ensure the validity of data collection tool. The data collectors were trained before the process of data collection. Supervision and checking was made by the supervisor to ensure the completeness and consistency of the collected data. All collected data were examined for completeness and consistency during data management, storage, and analysis.

## Data analysis and presentation

The collected data were entered into Epi Info 7 and analyzed using Statistical Package for Social Sciences (SPSS) version 20. Descriptive statistics like mean, frequency, and percentage were used to describe the characteristic of participants using table and text. multicollinearity among selected independent variables was checked through the variance inflation factor (VIF). Both binary and multivariate logistic regression analysis were done to identify determinants of poor glycemic control. In bivariate logistic regression analysis, variables with P-value

less than or equal to 0.2 were entered to multivariate logistic regression analysis to control for potential confounding variables that affect the poor glycemic level. Finally, statistically significant association of variables has been claimed based on the Adjusted Odds Ratio (AOR) with its 95%CI and P-value <0.05.

## Ethical consideration

Ethical approval was obtained from the Ethical Research Committee of school of pharmacy, department of clinical pharmacy, University of Gondar. A permission letter was obtained from Dilchora Referral Hospital to undertake the study. Written informed consent was obtained from each participant. Confidentiality of the information was assured and the privacy of the participant's medical record was maintained. To ensure confidentiality, names of patients and health care professionals were not recorded on the data collection tool.

## Operational definitions

**Good blood glucose control.** When the average fasting blood sugar level on the previous three consecutive occasions of their visit to a hospital is less than 154mg/dl [30].

**Poor blood glucose control.** When the average fasting blood sugar level on the previous three consecutive occasions of their visit to the a hospital is greater than or equal to 154mg/dl [30].

## Results

### Socio demographic and clinical characteristics of T2DM patients

A total of 422 patients were recruited to participate in this study. Of whom, 394 participants responded to the interview completely and included in the final analysis. The mean age of participants was 40.76 years with standard deviation (SD) of 12.79. More than half of the respondents 204 (51.8%) were females. Two hundred sixty (66%) participants lived in the urban areas. Most of the study participants 277 (70.3%) were married (Table 1). Half of the respondents 199(50.5%) had no family history of DM. The overall mean of the duration of DM since diagnosis was 8.93 ±5.67 years, with a minimum of 1 years and maximum of 30 years. More than two third of respondents 277 (70.3%) had no comorbid disease. About 166 (42.1%) respondents had diabetic complications. Among the study participants, 28(7.1%) and 46 (11.7%) respondents took atorvastatin and enalapril for the treatment of dyslipidemia and hypertension, respectively. The overall mean of the duration of DM treatment was 8.34 ±5.6 SD years, with a minimum of 1 year and a maximum of 30 years. Out of the total participants, 180(45.7%) respondents were taking insulin alone (Table 2).

### Interaction of T2DM patients with pharmacist

Around 171(43.4%) respondents had a poor interaction with pharmacists. Nearly sixty percent of respondents 243(61.7%) preferred the Amharic language to communicate with a pharmacist. More than half of the study participants 233(59.1%) understood the pharmacist's advice regarding their medication. More than fifty percent of respondents 208(52.8%) were not satisfied with the overall pharmaceutical service obtained from pharmacists (Table 3).

### Knowledge, attitude and practice of T2DM patients

More than half of the respondents 222(56.3%) had inadequate knowledge about T2DM. Around 288 (73.1%) respondents had a good attitude towards T2DM. Less than half of the respondents190 (48.2%) had a poor practice of DM (Table 4).

**Table 1. Socio demographic characteristics of T2DM patients who were attending Dilchora Referral Hospital, September 2019.**

| Characteristics | | Frequency (%) | Glycemic level | |
|---|---|---|---|---|
| | | | Good (%) | Poor (%) |
| Sex | Female | 204(51.8) | 104(51) | 100(49) |
| | Male | 190(48.2) | 112(58.9) | 78(41.1)) |
| Age (years) | 18–39 | 175(44.4) | 94(53.7) | 81(46.3) |
| | 40–59 | 184(46.7) | 101(54.9) | 83(45.1) |
| | ≥ 60 | 35(8.9) | 21(60) | 14(40) |
| Current residence | Urban | 260(66) | 141(54.2) | 119(45.8) |
| | Rural | 134(34) | 75(56) | 59(44) |
| Educational level | Unable to read and write | 79(20.1) | 52(65.8) | 27(34.2) |
| | Able to read and write | 48(12.2) | 23(47.9) | 25(52.1) |
| | Primary school | 86(21.8) | 43(50) | 43(50) |
| | Secondary school | 104(26.4) | 58(55.8) | 46(44.2) |
| | Tertiary and above | 77(19.5) | 40(51.9) | 37(48.1) |
| Marital status | Single | 81(20.6) | 39(48.1) | 42(51.9) |
| | Married | 277(70.3) | 159(57.4) | 118(42.6) |
| | Widowed | 6(1.5) | 3(50) | 3(50) |
| | Divorced | 30(7.6) | 15(50) | 15(50) |
| Occupation | Student | 69(17.5) | 33(47.8) | 36(52.2) |
| | Employed | 131(33.2) | 80(61.1) | 51(38.9) |
| | Housewife | 64(16.2) | 33(51.6) | 31(48.1) |
| | Merchant | 52(13.2) | 30(57.7) | 22(42.3) |
| | Daily laborer | 78(19.8) | 40(51.3) | 38(48.7) |
| Religion | Orthodox | 180(45.7) | 97(53.9) | 83(46.1) |
| | Protestant | 70(17.8) | 39(55.7) | 31(44.3) |
| | Muslim | 144(36.5) | 80(55.6) | 64(44.4) |
| Ethnicity | Oromo | 148(37.6) | 76(51.4) | 72(48.6) |
| | Amhara | 104(26.4) | 57(54.8) | 47(45.2) |
| | Somali | 50(12.7) | 30(60) | 20(40) |
| | Tigrae | 54(13.7) | 32(59.3) | 22(40.7) |
| | Woleyta | 38(9.6) | 21(55.3) | 17(44.7) |

## Prevalence of poor glycemic control in T2DM patients

The overall prevalence of poor glycemic control was 45.2% (95%CI: 40.6–50.0). The overall mean (SD) fasting blood sugar was 154.57mg/dl ± 36.33 SD. Around 100(56.2%) females, 119 (66.85) respondents who live in urban and a quarter of subjects in primary 43(24.15%) and secondary school 46(25.8%) had poor glycemic control. The majority of respondents who had poor glycemic control were married 118(66.3%). Approximately fifty percent of participants 92(51.7%) who had no family history of DM had poor glycemic control. Ninety-nine (55.6%) subjects had poor glycemic control without DM complications. Poor glycemic control was predominant in the participants 126(70.8%) who had no comorbid disease. Around 79(44.4%) respondents who had poor rates of interaction with a pharmacist had poor glycemic control. Near to sixty percent of participants 113(63.5%) who understood the advice of pharmacist about their drugs had poor glycemic control. More than fifty percent 103(57.9%) of participants with inadequate knowledge had poor glycemic control. The prevalence of poor glycemic control was found to be 132(74.2%) and 98(55.1%) in patients with good attitude and practice, respectively. The prevalence of poor glycemic control among participants who had been taking

**Table 2. Clinical characteristics of T2DM patients who were attending Dilchora Hospital, September 2019.**

| Characteristics | | Frequency (%) | Glycemic level | |
|---|---|---|---|---|
| | | | Good (%) | Poor (%) |
| Family history of DM | Yes | 195(49.5) | 109(55.9) | 86(44.1) |
| | No | 199(50.5) | 107(53.8) | 92(46.2) |
| Duration of DM | < 7 years | 162(41.1) | 91(56.2) | 71(43.8) |
| | ≥ 7 years | 232(58.9) | 125(53.9) | 107(47.1) |
| Comorbid disease | Yes | 117(29.7) | 65(55.6) | 52(44.4) |
| | No | 277(70.3) | 151(54.5) | 126(45.5) |
| Diabetic complication | Yes | 166(42.1) | 87(52.4) | 79(47.6) |
| | No | 228(57.9) | 129(56.6) | 99(43.4) |
| Drug given for dyslipidemia | Atorvastatin | 28(7.1) | 16(57.1) | 12(42.9) |
| | Simvastatin | 18(4.6) | 10(55.6) | 8(44.4) |
| Drug given for hypertension | Amlodipine | 11(2.8) | 11(100) | 0(0.0) |
| | Enalapril | 46(11.7) | 23(50) | 23(50) |
| | Enalapril + Amlodipine | 8(2) | 5(62.5) | 3(37.5) |
| | Enalapril + HCT | 11(2.8) | 3(27.3) | 8(72.7) |
| | Enalapril + HCT + Amlodipine | 1(0.3) | 1(100) | 0(0.0) |
| | Enalapril + Nifedipine | 3(0.8) | 2(66.7) | 1(33.3) |
| | HCT+ Amlodipine | 4(1) | 2(50) | 2(50) |
| | HCT + Nifedipine | 1(0.3) | 1(100) | 0(0.0) |
| | HCT | 4(1) | 3(75) | 1(25) |
| | Nifedipine | 4(1) | 2(50) | 2(50) |
| Duration of DM treatment | < 7 years | 207(52.5) | 117(56.5) | 90(43.5) |
| | ≥ 7 years | 187(47.5) | 99(52.9) | 88(47.1) |
| Total number of anti-diabetic drugs | One drug | 298(75.6) | 170(57) | 128(43) |
| | Two drugs | 88(22.3) | 41(46.6) | 47(53.4) |
| | Three drugs | 8(2) | 5(62.5) | 3(37.5) |
| Types of treatment | Oral anti diabetic drug | 164(41.6) | 92(56.1) | 72(43.9) |
| | Insulin | 180(45.7) | 105(58.3) | 75(41.7) |
| | Oral anti diabetics + insulin | 50(12.7) | 19(38) | 31(62) |

**Table 3. Interaction of T2DM patients who were attending Dilchora Referral Hospital with Pharmacist, September 2019.**

| *Characteristics* | | Frequency (%) | Glycemic level | |
|---|---|---|---|---|
| | | | Good (%) | Poor (%) |
| rate of interaction with the pharmacist | Good | 141(35.8) | 80(56.7) | 61(43.3) |
| | Moderate | 82(20.8) | 44(53.7) | 38(46.3) |
| | Poor | 171(43.4) | 92(53.8) | 79(46.2) |
| Patient language preference to communicate with pharmacist | Amharic | 243(61.7) | 131(53.9) | 112(46.1) |
| | Afan oromo | 70(17.8) | 42(60) | 28(40) |
| | Adarigna | 3(0.8) | 3(100) | 0(0.0) |
| | Somali | 33(8.4) | 18(54.5) | 15(45.5) |
| | Woleytgna | 45(11.4) | 22(48.9) | 23(51.1) |
| clarity of the pharmacist advice about their drug | clear | 233(59.1) | 120(51.5) | 113(48.5) |
| | not clear | 161(40.9) | 96(59.6) | 65(40.4) |
| Satisfied with overall pharmaceutical service get from pharmacists | Yes | 186(47.2) | 102(54.8) | 84(45.2) |
| | No | 208(52.8) | 114(54.8) | 94(45.2) |

**Table 4. Knowledge, attitude and practice of T2DM patients who were attending in Dilchora Hospital, September 2019.**

| Characteristics | | Frequency (%) | Glycemic level | |
|---|---|---|---|---|
| | | | Good (%) | Poor (%) |
| Level of Knowledge | Adequate knowledge | 172(43.7) | 97(56.4) | 75(43.6) |
| | Inadequate knowledge | 222(56.3) | 119(53.6) | 103(46.4) |
| Level of Attitude | Good attitude | 288(73.1) | 156(54.2) | 132(45.8) |
| | Poor attitude | 106(26.9) | 60(56.6) | 46(43.4) |
| Level of Practice | Good practice | 204(51.8) | 124(60.8) | 80(39.2) |
| | Poor practice | 190(48.2) | 92(48.4) | 98(51.6) |

oral anti-diabetic drugs was 72(40.4%) while among those who had been taking insulin was 75 (42.1%) (Tables 1–4).

## Factors associated with poor glycemic control among T2DM patients

In multivariable logistic regression analysis, the variables with significant effects on poor glycemic control include taking an oral anti-diabetic drug with insulin, level of understanding of pharmacist' advice regarding drugs and poor practice of diabetic patients. Participants who were on oral anti-diabetic drug plus insulin had nearly two times the likelihood of poor glycemic control than those who were on oral anti-diabetic drug alone: adjusted odds ratio (AOR) = 2.177; 95% confidence interval (CI): [1.104, 4.294; p = 0.025]. The odd of poor glycemic control in participants who were not able to understand the pharmacist's advice regarding their drug was approximately two times higher than in those who understood the pharmacist's advice: AOR = 1.857; 95%CI: [1.100, 3.132; p = 0.020]. patients with a poor level of practice were 1.5 times more likely to have poor glycemic control than those who had a good level of practice: AOR = 1.693; 95% CI: [1.126, 2.545; p = 0.011] (Table 5).

## Discussion

This study was carried out to assess poor glycemic control and its determinants among T2DM outpatients in one of the major hospitals in Eastern Ethiopia. Poor glycemic control was observed in 45.2% of participants (95%CI: 40.6–50.0). Being on oral anti-diabetic drug plus insulin therapy, unable to understood pharmacist's advice about their drug and having a poor practice of DM were significantly associated with poor glycemic control.

**Table 5. Multivariable analysis factors associated with poor glycemic control among T2DM patients Dilchora Hospital, September 2019.**

| Variables | | Glycemic level | | COR (95%CI) | AOR (95%CI) | P value |
|---|---|---|---|---|---|---|
| | | Good (%) | Poor (%) | | | |
| Types of treatment | OAD | 92(56.1) | 72(43.9) | 1 | 1 | 1 |
| | Insulin | 105(58.3) | 75(41.7) | 0.913(0.595–1.400) | 0.87(0.54–1.37) | 0.53 |
| | OAD + insulin | 19(38) | 31(62) | 2.085(1.09–3.990) | 2.177(1.104–4.294) | 0.025 |
| Sex | Female | 104(51) | 100(49) | 1 | 1 | |
| | Male | 112(58.9) | 78(41.1) | 0.72(0.49–1.1) | 1.41(0.89–2.2) | 0.136 |
| Clarity of Pharmacist's advice about drug | Clear | 120(51.5) | 113(48.5) | 1 | | 1 |
| | Not clear | 96(59.6) | 65(40.4) | 0.72(0.5–0.9) | 1.857(1.100–3.132) | 0.020 |
| Level of Practice | Good practice | 124(60.8) | 80(39.2) | 1 | 1 | 1 |
| | Poor practice | 92(48.4) | 98(51.6) | 1.651(1.107–2.463) | 1.693(1.126–2.545) | 0.011 |

OAD: Oral Anti diabetic Drugs.

In the present study, 45.2% of the participants had poor glycemic control. The proportion of poor glycemic control was comparable to the results reported in Ambo, Brazil, Iran, and Jordan [11, 17, 31, 32]. In other studies, carried out in Riyadh (67.7%), Al-Hasa (67.9%), Jazan (74%), Oman (65.0%), United Arab Emirates (69%), Kuwait (78.8%) and Rawalpindi (76%) [33–39], poor glycemic control was higher unlike the current study. This discrepancy may have happened due to the difference in socioeconomic status, culture, environmental factors, and lifestyle, which predispose individuals to different risk factors of poor glycemic control.

In this study, a high-level prevalence of poor glycemic control was presented among study participants who were on insulin treatment (40%) and Oral anti–diabetic drugs (42.1%). The results were lined with other studies [20, 40]. Starting insulin therapy for T2DM patients often showed blood glucose level is not well controlled [41, 42].

In the current study, a higher proportion of poor glycemic control was reported in study participants who live with DM for a long duration of period since diagnosis (60.1%). But a study conducted at the University of Gondar Hospital revealed that a high proportion of poor glycemic control was observed in those patients who live with DM for less than seven years (68.5%) [43]. It is understood that progressive impairment of insulin secretion through time because of β cell failure could lead to poor glycemic control [44].

This study revealed that the combination of an oral anti-diabetic drug plus insulin therapy is significantly associated with a poor glycemic control. The finding is in line with prior research studies [20, 45] and contradicts the study conducted in India which reported no significant association between oral anti-diabetic drug plus insulin therapy with poor glycemic control [46].

In the current study, participants who had a poor level of practice were found to have poor glycemic control than those who had a good level of practice. Because of the progressive nature of T2DM, treatment with drugs alone is not adequate to maintain euglycemia over time. Rather, after the medication is initiated, diabetic patients are encouraged to include lifestyle management including avoiding refined sugars as in soft drinks, increase in the amount of fiber, avoiding cigarettes, other tobacco products and alcohol and engaging in regular aerobic activity [1, 47]. The odd of poor glycemic control in study participants who were not able to understand the pharmacist's advice was higher than in those who understand the pharmacist advice. An interventional study conducted in Nadu and Iraq revealed a statistically significant reduction in the mean blood glucose level among patients advised by pharmacists appropriately [48, 49]. Patient-pharmacist interaction might improve patient's adherence to medication and other instructions which in turn help to achieve adequate control of DM. In this study factors associated with poor glycemic control were assessed using across-sectional design, which might not show causal relationships with potential risk factors.

## Conclusion and recommendation

This study revealed that the overall prevalence of poor glycemic control was high in Dilchora Referral Hospital. Patients on a combination of hypoglycemic drugs and insulin, a poor understanding of pharmacist's advice regarding medications and having poor practice of T2DM were risk factors for poor glycemic control. Health professionals working in the hospital should provide better patient advice about medications and should design treatment strategies for T2DM. A cohort study is recommended to infer substantial evidence of causality. In addition, the level of glycemic control should be determined by using the HbA1c test which is a good predictor of glycemic control over a long period of time.

## Supporting information

**S1 File.**
(DOCX)

**S2 File.**
(DOCX)

## Acknowledgments

The authors would like to express gratitude to the participants of the study for their time and the information they provided. We would also like to acknowledge the administrators of Dilchora Referral Hospital and the data collectors for their cooperation. Lastly, we would like to thank our colleague for reviewing grammar.

## Author Contributions

**Conceptualization:** Shambel Nigussie, Nigussie Birhan, Firehiwot Amare, Getnet Mengistu, Fuad Adem, Tadesse Melaku Abegaz.

**Data curation:** Tadesse Melaku Abegaz.

**Formal analysis:** Shambel Nigussie, Nigussie Birhan, Firehiwot Amare, Getnet Mengistu, Fuad Adem, Tadesse Melaku Abegaz.

**Methodology:** Shambel Nigussie, Nigussie Birhan, Firehiwot Amare, Getnet Mengistu, Fuad Adem, Tadesse Melaku Abegaz.

**Supervision:** Shambel Nigussie.

**Validation:** Tadesse Melaku Abegaz.

**Visualization:** Tadesse Melaku Abegaz.

**Writing – original draft:** Shambel Nigussie, Nigussie Birhan, Firehiwot Amare, Getnet Mengistu, Fuad Adem, Tadesse Melaku Abegaz.

**Writing – review & editing:** Shambel Nigussie, Nigussie Birhan, Firehiwot Amare, Getnet Mengistu, Fuad Adem, Tadesse Melaku Abegaz.

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
