## [Decision Letter · Decision Letter 0]

6 Apr 2021

PONE-D-20-38942

Rate of glycemic control and associated factors among type two Diabetes mellitus patients in Ethiopia: A cross sectional study

PLOS ONE

Dear Dr. Nigussie,

Thank you for submitting your manuscript to PLOS ONE. After careful consideration, we feel that it has merit but does not fully meet PLOS ONE’s publication criteria as it currently stands. Therefore, we invite you to submit a revised version of the manuscript that addresses the points raised during the review process.

The reviewers have criticized the title, methodology and selection criteria of the study. You are invited to revise your manuscript keeping in view that your revision will be re-evaluated by the reviewers. Moreover, when you revise the manuscript, pay special intention to remove all the possible grammatical mistakes and syntax errors.

We look forward to receiving your revised manuscript.

Kind regards,

Muhammad Sajid Hamid Akash

Academic Editor

PLOS ONE

Journal Requirements:

1. Please ensure that your manuscript meets PLOS ONE's style requirements, including those for file naming. The PLOS ONE style templates can be found athttps://journals.plos.org/plosone/s/file?id=wjVg/PLOSOne_formatting_sample_main_body.pdf and https://journals.plos.org/plosone/s/file?id=ba62/PLOSOne_formatting_sample_title_authors_affiliations.pdf

- https://www.ncbi.nlm.nih.gov/pmc/articles/PMC4315535/?tool=pmcentrez

- https://jimdc.org.pk/jimdc/Volumes/4-2/Factors%20Associated%20with%20Uncontrolled%20Type%202%20Diabetes%20Mellitus.pdf

The text that needs to be addressed involves paragraph 3 of the Discussion section.

In your revision ensure you cite all your sources (including your own works), and quote or rephrase any duplicated text outside the methods section. Further consideration is dependent on these concerns being addressed.

3. In the ethics statement in the Methods section and online submission information, please specify the type of informed consent that was obtained from the participants (for instance, written or verbal, and if verbal, how it was documented and witnessed).

"Please include additional information regarding the survey or questionnaire used in the study and ensure that you have provided sufficient details that others could replicate the analyses. For instance, if you developed a questionnaire as part of this study and it is not under a copyright more restrictive than CC-BY, please include a copy, in both the original language and English, as Supporting Information, or include a citation if it has been published previously.”

"In the Methods, please discuss whether and how the questionnaire was validated and/or pre-tested. If these did not occur, please provide the rationale for not doing so."

"To comply with PLOS ONE submission guidelines, in your Methods section, please provide additional information regarding your statistical analyses. For more information on PLOS ONE's expectations for statistical reporting, please see https://journals.plos.org/plosone/s/submission-guidelines.#loc-statistical-reporting.

Additional Editor Comments (if provided):

Reviewers' comments:

Reviewer's Responses to Questions

**Comments to the Author**

1. Is the manuscript technically sound, and do the data support the conclusions?

Reviewer #1: Partly

Reviewer #2: Yes

2. Has the statistical analysis been performed appropriately and rigorously? 

Reviewer #1: No

Reviewer #2: Yes

3. Have the authors made all data underlying the findings in their manuscript fully available?

Reviewer #1: Yes

Reviewer #2: Yes

4. Is the manuscript presented in an intelligible fashion and written in standard English?

Reviewer #1: No

Reviewer #2: Yes

5. Review Comments to the Author

Reviewer #1: It's an interesting study, however following corrections are needed before further process

Title

Type Two Diabetes Mellitus in title and key words should corrected as " Type 2 Diabetes Mellitus"

Introduction

Study pattern should also rationalized rather than writing about previous studies

Methodology

How the inter rater reliability was ensured during data collection?

For structured questionnaire , how the pool of questions was finalized.

How the reliability was assessed?

How the validity was ensured ?

Did this tools can be applied as self administered ?

What were the factors to include marital status, occupation, religion in questionnaire.

In the selection of sample and sample size, a prospective method is used, Why the sample size is calculated by using a general population formula. Please elaborate and justify.

Results

What is the correlation of T2D with hypertension ?

What is the correlation demographics with T2D?

Impact of co morbidity on T2D control

Discussion

Discussion with illustration will make this study more worthy.

Conclusion

Should be on the basis of results obtained

Reviewer #2: This paper is well written and with an excellent rationale of study. The following comments must be addressed.

Elaborate the source of the study population in terms of available data from the site of study (attach relevant annexures if required).

The sample size 394 is not justifiable based on the published literature of the 500,000 population of the area you have carried research. How can you claim that the sample size 394 is representative of the said population? Why was not convenience sampling used?

The Danial Sample size formula used here needs justification.

Share the results of the pretest done on 5% of the population for quality test.

How will you infer the temporal association between a risk factor and the outcome of your research disease?

The references in this study are quite outdated.

Add some latest studies in the discussion section.

State in the recommendations that how the results of this study may inform the hypotheses for a more complex investigation, such as a cohort study.

6. PLOS authors have the option to publish the peer review history of their article (what does this mean?). If published, this will include your full peer review and any attached files.

Reviewer #1: **Yes: **Muhammad Majid Aziz

Reviewer #2: No

---

## [Author Response · Author response to Decision Letter 0]

23 Apr 2021

Authors response to reviewers

Title: Rate of Glycemic Control and Associated Factors Among Type Two Diabetes Mellitus Patients in Ethiopia: A Cross Sectional Study 

Manuscript reference: [PONE-D-20-38942]- [EMID: d55568cd7be1bbfe]

Subject: Revision of the manuscript

We thank the journal and the editors for allowing as the opportunity to publish our work in well reputed journal. We would like to thank the reviewers for the worthy comments. The comments have indeed helped us improve the quality of the paper. We revised the paper extensively and respond in detail to the question and comments for the reviewers. We are open to any further improvements in the paper. Find below the point- by- point description of changes.

For Academic editor

Comment 1: Please ensure that your manuscript meets PLOS ONE's style requirements, including those for file naming

Response 1: thank you for your comment. Now the revised version is fulfilled style requirements.

Comment 2: We noticed you have some minor occurrence of overlapping text with the following previous publication(s), which needs to be addressed:

- https://www.ncbi.nlm.nih.gov/pmc/articles/PMC4315535/?tool=pmcentrez

- https://jimdc.org.pk/jimdc/Volumes/4-2/Factors%20Associated%20with%20Uncontrolled%20Type%202%20Diabetes%20Mellitus.pdf

 Response 2: Thank you for your constructive comment. This comment incorporated in the overall of main document.

Comment 3: In the ethics statement in the Methods section and online submission information, please specify the type of informed consent that was obtained from the participants (for instance, written or verbal, and if verbal, how it was documented and witnessed.

Response 3. Thank you for your comment. Written informed consent was used. The data collectors read the written consent to participants and if the they agreed, signed on it.

Comment 4: "Please include additional information regarding the survey or questionnaire used in the study and ensure that you have provided sufficient details that others could replicate the analyses. For instance, if you developed a questionnaire as part of this study and it is not under a copyright more restrictive than CC-BY, please include a copy, in both the original language and English, as Supporting Information, or include a citation if it has been published previously.”

Response 4: Thank you for your comment. we attached the questionnaire in supporting information 

Comment 5: "In the Methods, please discuss whether and how the questionnaire was validated and/or pre-tested. If these did not occur, please provide the rationale for not doing so."

Response 5: Thank you for your comment. We incorporated in the new version of main document.

Comment 6: To comply with PLOS ONE submission guidelines, in your Methods section, please provide additional information regarding your statistical analyses.

Response 6: Thank you for your constructive comment. we included this comment in new version of main document.

Comment 7: Please ensure that you have an ORCID iD and that it is validated in Editorial Manager.

 Response 7: Thank you for your specific comment. I have validated my ORCID ID. 

Comment 8: Please include captions for your Supporting Information files at the end of your manuscript, and update any in-text citations to match accordingly.

 Response 8: Thank you for your comment. The comment considered during submission of revised version.

For reviewer 1

 On Title 

Comment 1: Correct title and key word as follow: Type Two Diabetes Mellitus in title and key words should corrected as " Type 2 Diabetes Mellitus"

 Response 1: Thank you for your specific comment. This comment incorporated in revised version.

 On Introduction

Comment 1: Study pattern should also rationalize rather than writing about previous studies.

Response 1: We accept your comment; the comment is incorporated into the revised document. 

 On Methodology

Comment 1: How the inter rater reliability was ensured during data collection? 

Response1: Thank you for your comment. The data collectors were given code for each participant questionnaires at the time of data collection and then before submitting the collected data to supervisor; the data collector submitted collected data to their colleague to check any error and to have common understanding for those not understand during training.

Comment 2: How the reliability was assessed?

Response 2: pretest was done on 22 randomly selected T2DM patients to ensure consistency of the data abstraction format and the structured questionnaire. Any error found during the process of the pretest was corrected and modification was made into the final version of the data abstraction format and the structured questionnaire.

Comment 3: How the validity was ensured?

Response 3: Thank you for your comment. After we developing the questionnaire by reviewing different literature; we submitted to the experts to comment our questionnaire and by incorporating their comment finalized our questionnaire.

Comment 4: Did this tool can be applied as self-administered? 

Response 4: Thank you for your constructive comment. Yes, but we were not applied as self-administered; because some of our study participants were can’t read and write 

Comment 5: What were the factors to include marital status, occupation, religion in questionnaire.

Response 5: Thank you for your nice comment: one study pointed out marital status as one of contributing factor for poor glycemic control. Even if no study showed that occupation and religion as contributing factor for poor glycemic control, experts recommended to study these two variables. Because fasting in religion and having no any job in occupation are risk factor for poor glycemic control.

Comment 6: In the selection of sample and sample size, a prospective method is used, Why the sample size is calculated by using a general population formula. Please elaborate and justify.

Responses 6: thank you for your comment. The total population what we want to generalize the results were high. Finally, we decided to calculate largest sample size that represent the total population by using a general population formula. 

 On Result

Comment 1: What is the correlation of T2D with hypertension?

Response 1: Thank you for your comment. In this study there is no significant association between hypertension and T2DM. 

Comment 2: What is the correlation demographics with T2D?

Response 2: Thank you for your comment. In this study there is so significant association between demographic and T2DM. 

Comment 3: Impact of co morbidity on T2D control

Response 3: Thank you for your comment. In this study there is no significant association between comorbidity and T2DM.

 On Discussion part

Comment 1: Discussion with illustration will make this study more worthy

Response 1: Thank you for your constructive comment. We incorporated in the new version of main document

 On Conclusion part

Comment 1: conclusion should be on the basis of results obtained 

Response 1: Thank you for your nice comment. We incorporated in modified version of main document. 

For reviewer 2

Comment 1: Elaborate the source of the study population in terms of available data from the site of study

Response 1: Thank you for your constructive comment. The source population for this study was all diabetes mellitus patients who were registered for follow-up at diabetic clinic of Dilchora referral Hospital with total number of 1768.

Comment 2: The sample size 394 is not justifiable based on the published literature of the 500,000 population of the area you have carried research. How can you claim that the sample size 394 is representative of the said population? 

Response 2: Thank you for your comment. The required sample size is determined by using a formula for single population proportion by taking different P value from different previous studies and the largest calculated sample size was taken that represent targeted population. 

Comment 3: Why was not convenience sampling used?

 Response 3: Thank you for your constructive comment. We used systematic random sampling rather than convenience sampling. Because the result of convenience sampling lacks generalizability on our target population.

Comment 4: The Danial Sample size formula used here needs justification.

Response 4: Thank you for your comment. We incorporated the justification in the new version of main document.

Comment 5: Share the results of the pretest done on 5% of the population for quality test.

Response 5: Thank you for your comment. We attached the result of pretest in supporting information.

Comment 6: How will you infer the temporal association between a risk factor and the outcome of your research disease? 

Response 6: Thank you for your constructive comment. Our study design was cross-sectional, which might not show causal relationships with potential risk factors

Comment 7: The references in this study are quite outdated.

Response 7: Thank you for your comment. We replaced some of outdated reference with its updated reference in the new version of main document.

Comment 8: Add some latest studies in the discussion section.

Response 8: Thank you for your nice comment. We incorporated recent studies in the new version of main document.

Comment 9: State in the recommendations that how the results of this study may inform the hypotheses for a more complex investigation, such as a cohort study.

Response 9: Thank you for your comment. We incorporated the recommendation in the new version of main document

---

## [Editor Report · Decision Letter 1]

28 Apr 2021

Rate of glycemic control and associated factors among Type Two Diabetes Mellitus patients in Ethiopia: A cross sectional study

PONE-D-20-38942R1

Dear Dr. Nigussie,

We’re pleased to inform you that your manuscript has been judged scientifically suitable for publication and will be formally accepted for publication once it meets all outstanding technical requirements.

Kind regards,

Muhammad Sajid Hamid Akash

Academic Editor

PLOS ONE
---

## [Editor Report · Acceptance letter]

30 Apr 2021

PONE-D-20-38942R1 

Rate of Glycemic Control and Associated Factors Among Type Two Diabetes Mellitus Patients in Ethiopia: A Cross Sectional Study  

Dear Dr. Nigussie:

I'm pleased to inform you that your manuscript has been deemed suitable for publication in PLOS ONE. Congratulations! Your manuscript is now with our production department. 

Kind regards, 

on behalf of

Dr. Muhammad Sajid Hamid Akash 

Academic Editor

PLOS ONE